# Mixed effects analysis of factors associated with barriers to accessing healthcare among women in sub-Saharan Africa: Insights from demographic and health surveys

Abdul-Aziz Seidu [1,2] *

**1** Department of Population and Health, College of Humanities and Legal Studies, University of Cape Coast, Cape Coast, Ghana, **2** College of Public Health, Medical and Veterinary Sciences, James Cook University, Townsville, Queensland, Australia

* abdul-aziz.seidu@stu.ucc.edu.gh

## Abstract

### Background

Access to healthcare is one of the key global concerns as treasured in the Sustainable Development Goals. This study, therefore, sought to assess the individual and contextual factors associated with barriers to accessing healthcare among women in sub-Saharan Africa (SSA).

### Materials and methods

Data for this study were obtained from the latest Demographic and Health Surveys (DHS) conducted between January 2010 and December 2018 across 24 countries in SSA. The sample comprised 307,611 women aged 15–49. Data were analysed with STATA version 14.2 using both descriptive and multilevel logistic regression modelling. Statistical significance was set at p<0.05.

### Results

It was found that 61.5% of women in SSA face barriers in accessing healthcare. The predominant barriers were getting money needed for treatment (50.1%) and distance to health facility (37.3%). Women aged 35–39 (AOR = 0.945, CI: 0.911–0.980), married women (AOR = 0.694, CI: 0.658–0.732), richest women (AOR = 0.457, CI:0.443–0.472), and those who read newspaper or magazine at least once a week (AOR = 0.893, CI:0.811–0.983) had lower odds of facing barriers in accessing healthcare. However, those with no formal education (AOR = 1.803, CI:1.718–1.891), those in manual occupations (AOR = 1.551, CI: 1.424–1.689), those with parity 4 or more (AOR = 1.211, CI: 1.169–1.255), those who were not covered by health insurance (AOR = 1.284, CI: 1.248–1.322), and those in rural areas (AOR = 1.235, CI:1.209–1.26) had higher odds of facing barriers to healthcare access.

**Data Availability Statement:** The dataset is freely available for download at: https://dhsprogram.com/data/available-datasets.cfm. Survey years are included in Table 1.

**Funding:** The author(s) received no specific funding for this work.

**Competing interests:** The authors have declared that no competing interests exist.

**Abbreviations:** AOR, Adjusted Odds Ratio; CI, Confidence Interval; DHS, Demographic and Health Survey; VIF, Variance Inflation Factor; SDG, Sustainable Development Goals; LMICs, Low- and Middle-Income Countries; WHO, World Health Organisation; GSS, Ghana Statistical Service; LLR, log-likelihood ratio; SE, Standard Error; ICC, Intra-Class Correlation; LR Test, Likelihood ratio Test; AIC, Akaike's Information Criterion; BIC, Schwarz's Bayesian Information Criteria.

## Conclusion

Both individual and contextual factors are associated with barriers to healthcare accessibility in SSA. Particularly, age, marital status, employment, parity, health insurance coverage, exposure to mass media, wealth status and place of residence are associated with barriers to healthcare accessibility. These factors ought to be considered at the various countries in SSA to strengthen existing strategies and develop new interventions to help mitigate the barriers. Some of the SSA African countries can adopt successful programs in other parts of SSA to suit their context such as the National Health Insurance Scheme (NHIS) and the Community-based Health Planning and Services concepts in Ghana.

## Background

The health of women has been held in high esteem globally. This was prioritised by the erstwhile Millennium Development Goals (MDGs) and has been highlighted in its successor, the Sustainable Development Goals (SDGs) [1]. Specifically, SDG-3 targets 3.8 and 3.7 emphasize universal health coverage and access to sexual and reproductive healthcare services, including family planning information and education, and the integration of reproductive healthcare into national strategies and programmes by 2030 [2, 3]. SDG target 3.1 also aims at reducing maternal mortality to less than 70 per 100,000 live births by 2030 [4].

Despite the fact that these global targets have yielded positive results in terms of women's health outcomes, massive improvements are still needed. In 2016, 303,000 women died from maternal-related causes [3, 5] and sub-Saharan Africa (SSA) recorded over 60% of these deaths [6]. Furthermore, in almost all countries globally, non-communicable diseases have also been among the major causes of death and disability among women, with higher rates for low- and middle-income countries [7]. Evidence again suggests that most women face barriers in their quest to accessing healthcare, which has resulted in poorer health outcomes such as miscarriage, unsafe abortions, and stillbirths [8].

Country-level studies such as those from Ethiopia [9, 10], Rwanda [11], Cameroon, and India [12] have revealed that individual and contextual factors are likely to obstruct women's access to healthcare. Specifically, these barriers include transportation, geographical location, system organisational barriers, general availability of services, health information, waiting times and health infrastructure [13]. To achieve SDG 3, it is important to enhance universal access to health services that guarantee the health needs and aspirations of women of reproductive age. It is, therefore, prudent to get empirical evidence to provide a holistic understanding of the barriers to healthcare among women in SSA. This study, therefore, seeks to assess the individual and contextual factors associated with barriers in accessing healthcare among women in SSA.

## Materials and methods

### Data source

Data for this study were obtained from current Demographic and Health Surveys (DHS) conducted between January 1, 2010 and December 31, 2018 in 24 SSA countries (see Table 1). The choice of the 24 countries was influenced by the availability of the variables of interest in their datasets. DHS is a nationwide survey undertaken across low- and middle-income countries every five-year period[14]. The survey is representative of each of these countries and targets

**Table 1. Sample size.**

| Country | Survey Year | Weighted Sample | Weighted Percentage |
|---|---|---|---|
| 1. Benin | 2017–2018 | 15,410 | 5.0 |
| 2. Burundi | 2016–2017 | 16,783 | 5.5 |
| 3. Dr Congo | 2013–2014 | 18,667 | 6.1 |
| 4. Ethiopia | 2016 | 15,299 | 5.0 |
| 5. Gabon | 2012 | 8,213 | 2.7 |
| 6. Ghana | 2014 | 9,365 | 3.0 |
| 7. Gambia | 2013 | 10,051 | 3.3 |
| 8. Guinea | 2018 | 10,553 | 3.4 |
| 9. Kenya | 2014 | 14,501 | 4.7 |
| 10. Liberia | 2013 | 9,013 | 2.9 |
| 11. Lesotho | 2014 | 2,849 | 0.9 |
| 12. Mali | 2018 | 10,410 | 3.4 |
| 13. Malawi | 2015–2016 | 24,540 | 8.0 |
| 14. Nigeria | 2018 | 28,582 | 9.3 |
| 15. Niger | 2012 | 11,023 | 3.6 |
| 16. Namibia | 2013 | 9,100 | 3.0 |
| 17. Sierra Leone | 2013 | 16,350 | 5.3 |
| 18. Chad | 2014–2015 | 5,940 | 1.9 |
| 19. Togo | 2013–2014 | 9,381 | 3.1 |
| 20. Tanzania | 2015–2016 | 13,253 | 4.3 |
| 21. Uganda | 2016 | 18,458 | 6.0 |
| 22. South Africa | 2016 | 4,049 | 1.3 |
| 23. Zambia | 2018 | 16,014 | 5.2 |
| 24. Zimbabwe | 2015 | 9,809 | 3.2 |
| Total | - | **307,611** | 100.0 |

core maternal and child health indicators such as healthcare accessibility, unintended pregnancy, contraceptive use, skilled birth attendance, immunisation among under-fives, intimate partner violence, access to healthcare, and issues regarding men's health such as tobacco and contraceptive use. In selecting the sample for each survey, multi-stage sampling approach was employed. The first step of this sampling approach involved the selection of clusters (i.e., enumeration areas [EAs]), followed by systematic household sampling within the selected EAs. In this study, the sample size consisted of women aged 15–49 who had complete information on all the variables of interest (N = 307,611). The Strengthening Reporting of Observational studies in Epidemiology (STROBE) guideline was used in the preparation of this manuscript [15]. The dataset is freely available for download at https://dhsprogram.com/data/available-datasets.cfm

### Definition of variables

**Outcome variable.** The outcome variable in this study was barriers to healthcare accessibility. It was derived from four questions on barriers to healthcare access that each woman responded to. These focused on difficulty in obtaining money (money), distance to health facility (distance), getting permission for treatment (permission), and not wanting to go alone (companionship). If a woman faced at least one or more of the problems (money, distance, companionship, and permission), she was considered to have barriers to healthcare access and coded as "1". However, if she did not report difficulty in getting money, distance,

**Table 2. Socio-demographic characteristics and barriers to health care access among women in SSA.**

| Variables | Weighted Sample N = 307, 611 | | Barrier in Healthcare Access | | P-values |
|---|---|---|---|---|---|
| | n | % | No (%) | Yes (%) | |
| **Individual level factors** | | | | | |
| **Age** | | | | | p<0.001 |
| 15–19 | 61,599 | 20.0 | 39.0 | 61.0 | |
| 20–24 | 55,777 | 18.1 | 39.9 | 60.1 | |
| 25–29 | 54,677 | 17.8 | 40.0 | 60.0 | |
| 30–34 | 45,511 | 14.8 | 39.1 | 61.0 | |
| 35–39 | 38,719 | 12.6 | 38.3 | 61.7 | |
| 40–44 | 28,223 | 9.2 | 36.4 | 63.6 | |
| 45–49 | 23,106 | 7.5 | 34.7 | 65.3 | |
| **Marital status** | | | | | p<0.001 |
| Never married | 80,822 | 26.3 | 43.4 | 56.6 | |
| Married | 168,425 | 54.8 | 38.4 | 61.6 | |
| Cohabiting | 30,783 | 10.0 | 33.2 | 66.8 | |
| Widowed | 8,444 | 2.8 | 30.3 | 69.7 | |
| Divorced | 19,136 | 6.2 | 33.8 | 66.2 | |
| **Education** | | | | | p<0.001 |
| No education | 92,888 | 30.2 | 29.7 | 70.3 | |
| Primary | 99,495 | 32.3 | 33.2 | 66.9 | |
| Secondary | 98,832 | 32.1 | 47.8 | 52.2 | |
| Higher | 16,396 | 5.3 | 68.7 | 31.3 | |
| **Employment** | | | | | p<0.001 |
| Not working | 100,209 | 32.6 | 39.8 | 60.2 | |
| Managerial | 14,048 | 4.6 | 63.1 | 36.9 | |
| Clerical | 2,989 | 1.0 | 67.6 | 32.4 | |
| Sales | 56,511 | 18.4 | 44.6 | 55.4 | |
| House/domestic | 6,601 | 2.2 | 46.9 | 53.1 | |
| Agricultural | 78,344 | 25.5 | 25.6 | 74.4 | |
| Services | 22,872 | 7.4 | 43.0 | 57.0 | |
| Manual | 26,036 | 8.5 | 38.9 | 61.1 | |
| **Parity** | | | | | p<0.001 |
| None | 78,716 | 25.6 | 43.1 | 56.9 | |
| 1–3 children | 120,208 | 39.1 | 41.6 | 58.4 | |
| 4 or more children | 108,687 | 35.3 | 32.3 | 67.7 | |
| **Health insurance coverage** | | | | | p<0.001 |
| No | 281,465 | 91.5 | 37.2 | 62.8 | |
| Yes | 26,146 | 8.5 | 54.8 | 45.2 | |
| **Frequency of listening to radio** | | | | | p<0.001 |
| Not at all | 119,195 | 38.8 | 30.5 | 69.5 | |
| Less than once a week | 65,306 | 21.2 | 39.3 | 60.7 | |
| At least once a week | 113,563 | 36.9 | 46.3 | 53.8 | |
| Almost every day | 9,548 | 3.1 | 46.9 | 53.1 | |
| **Frequency of reading newspaper or magazine** | | | | | p<0.001 |
| Not at all | 238,357 | 77.5 | 34.8 | 65.3 | |
| Less than once a week | 37,626 | 12.2 | 49.9 | 50.1 | |

(*Continued*)

**Table 2.** (Continued)

| Variables | Weighted Sample N = 307, 611 | | Barrier in Healthcare Access | | P-values |
|---|---|---|---|---|---|
| | n | % | No (%) | Yes (%) | |
| **Individual level factors** | | | | | |
| At least once a week | 29,126 | 9.5 | 55.9 | 44.1 | |
| Almost every day | 2,503 | 0.8 | 46.5 | 53.6 | |
| **Frequency of watching television** | | | | | p<0.001 |
| Not at all | 180,039 | 58.5 | 29.3 | 70.7 | |
| Less than once a week | 41,284 | 13.4 | 43.9 | 56.1 | |
| At least once a week | 72,048 | 23.4 | 58.0 | 42.0 | |
| Almost every day | 14,240 | 4.6 | 45.2 | 54.8 | |
| **Contextual factors** | | | | | |
| **Sex of household head** | | | | | p<0.001 |
| Male | 221,333 | 72.0 | 38.5 | 61.5 | |
| Female | 86,278 | 28.1 | 39.2 | 60.8 | |
| **Wealth status** | | | | | p<0.001 |
| Poorest | 53,412 | 17.4 | 22.9 | 77.1 | |
| Poorer | 56,717 | 18.4 | 28.2 | 71.8 | |
| Middle | 59,132 | 19.2 | 34.3 | 65.8 | |
| Richer | 64,330 | 20.9 | 43.1 | 56.9 | |
| Richest | 74,021 | 24.1 | 57.8 | 42.2 | |
| **Type of place of residence** | | | | | p<0.001 |
| Urban | 116,585 | 37.9 | 51.7 | 48.3 | |
| Rural | 191,026 | 62.1 | 30.8 | 69.2 | |

companionship, and permission-related barriers, she was considered not to have barriers to healthcare access and coded as "0" [16–18].

**Independent variables.** Both individual and contextual level factors were considered in this study. These variables were chosen based on their statistically significant association with barriers to healthcare access in previous studies [16–18]. The individual level factors included age (15–19, 20–24, 25–29, 30–34, 35–39, 40–44, 45–49), marital status (never married, married, cohabiting, widowed, divorced), educational level (no education, primary, secondary, higher), employment (not working, managerial, clerical, sales, house/domestic, agricultural, services, manual), parity (0,1–3, 4 or more), health insurance subscription (yes, no), and exposure to mass media, specifically, radio, newspaper, and television (not at all, less than once a week, at least once a week, almost every day). The contextual variables were sex of household head (male, female), household wealth status (poorest, poorer, middle, richer, richest), and type of residence (urban, rural) (see Table 2).

## Statistical analyses

The data were analysed with STATA version 14.2 for MacOS. Three basic steps were followed to analyse the data. The first step was the use of descriptive statistics to describe the sample and also cross-tabulation of all the independent variables against barriers to healthcare access. The second step was a bivariate analysis to select potential variables for the regression analysis. Variables that were statistically significant at the bivariate analysis at p<0.05 were moved to the regression stage, which involved a two-level multilevel binary logistic regression analyses to

assess the individual and contextual factors associated with barriers to healthcare access. Clusters were considered as random effect to account for the unexplained variability at the community level [19]. Four models were fitted (see Table 3). Firstly, model I was an empty model and

Table 3.  Multilevel logistic regression of individual and contextual factors associated with barriers to healthcare among women in SSA.

| Variables | Model I | Model II AOR [95%CI] | Model III AOR [95%CI] | Model IV AOR [95%CI] |
|---|---|---|---|---|
| **Individual level factors** | | | | |
| **Age** | | | | |
| 15–19 | | 1.099*** | | 0.986 |
| | | [1.053–1.147] | | [0.944–1.030] |
| 20–24 | | 1.096*** | | 1.014 |
| | | [1.054–1.139] | | [0.975–1.055] |
| 25–29 | | 1.031 | | 0.986 |
| | | [0.994–1.069] | | [0.950–1.023] |
| 30–34 | | 0.972 | | 0.953** |
| | | [0.938–1.007] | | [0.920–0.988] |
| 35–39 | | 0.951** | | 0.945** |
| | | [0.917–0.986] | | [0.911–0.980] |
| 40–44 | | 0.978 | | 0.972 |
| | | [0.941–1.016] | | [0.935–1.011] |
| 45–49 | | Ref | | Ref |
| **Marital status** | | | | |
| Never married | | 0.836*** | | 0.834*** |
| | | [0.790–0.885] | | [0.787–0.883] |
| Married | | 0.690*** | | 0.694*** |
| | | [0.656–0.726] | | [0.658–0.732] |
| Cohabiting | | 0.958 | | 0.928* |
| | | [0.906–1.014] | | [0.876–0.984] |
| Widowed | | Ref | | Ref |
| Divorced | | 0.969 | | 0.966 |
| | | [0.913–1.028] | | [0.910–1.026] |
| **Education** | | | | |
| No education | | 2.265*** | | 1.803*** |
| | | [2.161–2.374] | | [1.718–1.891] |
| Primary | | 1.988*** | | 1.676*** |
| | | [1.900–2.080] | | [1.601–1.756] |
| Secondary | | 1.570*** | | 1.438*** |
| | | [1.504–1.638] | | [1.378–1.501] |
| Higher | | Ref | | Ref |
| **Employment** | | | | |
| Not working | | 1.552*** | | 1.449*** |
| | | [1.429–1.685] | | [1.334–1.573] |
| Managerial | | 1.186*** | | 1.156** |
| | | [1.086–1.295] | | [1.058–1.263] |
| Clerical | | Ref | | Ref |
| Sales | | 1.323*** | | 1.280*** |
| | | [1.218–1.438] | | [1.178–1.391] |
| House/domestic | | 1.403*** | | 1.409*** |

(*Continued*)

**Table 3.** (*Continued*)

| Variables | Model I | Model II<br>AOR [95%CI] | Model III<br>AOR [95%CI] | Model IV<br>AOR [95%CI] |
|---|---|---|---|---|
| | | [1.273–1.547] | | [1.278–1.553] |
| Agricultural | | 2.275*** | | 1.909*** |
| | | [2.093–2.473] | | [1.755–2.075] |
| Services | | 1.537*** | | 1.420*** |
| | | [1.411–1.674] | | [1.303–1.548] |
| Manual | | 1.588*** | | 1.551*** |
| | | [1.458–1.729] | | [1.424–1.689] |
| **Parity** | | | | |
| None | | Ref | | Ref |
| 1–3 | | 1.075*** | | 1.016 |
| | | [1.044–1.106] | | [0.987–1.046] |
| 4+ | | 1.343*** | | 1.211*** |
| | | [1.296–1.391] | | [1.169–1.255] |
| **Health insurance coverage** | | | | |
| No | | 1.251*** | | 1.284*** |
| | | [1.216–1.287] | | [1.248–1.322] |
| Yes | | Ref | | Ref |
| **Frequency of listening to radio** | | | | |
| Not at all | | 1.433*** | | 1.399*** |
| | | [1.365–1.506] | | [1.331–1.470] |
| Less than once a week | | 1.278*** | | 1.296*** |
| | | [1.215–1.344] | | [1.231–1.364] |
| At least once a week | | 1.155*** | | 1.174*** |
| | | [1.100–1.214] | | [1.117–1.234] |
| Almost every day | | Ref | | Ref |
| **Frequency of reading newspaper or magazine** | | | | |
| Not at all | | 1.034 | | 1.03 |
| | | [0.942–1.135] | | [0.937–1.132] |
| Less than once a week | | 0.893* | | 0.896* |
| | | [0.813–0.982] | | [0.814–0.986] |
| At least once a week | | 0.880** | | 0.893* |
| | | [0.800–0.968] | | [0.811–0.983] |
| Almost every day | | Ref | | Ref |
| **Frequency of watching television** | | | | |
| Not at all | | 1.151*** | | 0.907*** |
| | | [1.101–1.204] | | [0.866–0.950] |
| Less than once a week | | 0.778*** | | 0.706*** |
| | | [0.741–0.816] | | [0.673–0.742] |
| At least once a week | | 0.577*** | | 0.598*** |
| | | [0.551–0.604] | | [0.571–0.626] |
| Almost every day | | Ref | | Ref |
| **Contextual factors** | | | | |
| **Sex of household head** | | | | |
| Male | | | 0.990 | 1.000 |
| | | | [0.973–1.007] | [0.981–1.020] |
| Female | | | Ref | Ref |

(*Continued*)

**Table 3.** (Continued)

| Variables | Model I | Model II AOR [95%CI] | Model III AOR [95%CI] | Model IV AOR [95%CI] |
|---|---|---|---|---|
| **Wealth status** | | | | |
| Poorest | | | Ref | Ref |
| Poorer | | | 0.730*** | 0.785*** |
| | | | [0.710–0.749] | [0.764–0.806] |
| Middle | | | 0.567*** | 0.658*** |
| | | | [0.552–0.582] | [0.641–0.676] |
| Richer | | | 0.435*** | 0.570*** |
| | | | [0.424–0.447] | [0.554–0.586] |
| Richest | | | 0.274*** | 0.457*** |
| | | | [0.266–0.282] | [0.443–0.472] |
| **Place of residence** | | | | |
| Urban | | | Ref | Ref |
| Rural | | | 1.499*** | 1.235*** |
| | | | [1.458–1.262] | [1.209–1.262 |
| *N* | | 307,611 | 307,611 | 307,611 |
| **Parameters** | | | | |
| Community-level variance (SE) | 0.43(0.022) | 0.29(0.017) | 0.34(0.019) | 0.27(0.175) |
| ICC (%) | 11.7% | 8.1% | 9.6% | 8.2% |
| Log-likelihood | -201775.6 | -189152.7 | 191013.06 | -186503.04 |
| LR Test | 5866.45 (p<0.001) | 3772.84 (p<0.001) | 4522.01 (p<0.001) | 3839.23 (p<0.001) |
| AIC | 403555.2 | 378367.5 | 382042.1 | 373086.1 |
| BIC | 403576.5 | 378697.2 | 382127.2 | 373511.5 |

Exponentiated coefficients; 95% confidence intervals in brackets.

* $p < 0.05$

** $p < 0.01$

*** $p < 0.001$.

SE = Standard Error; ICC = Intra-Class Correlation; LR Test = Likelihood ratio Test; AIC = Akaike's Information Criterion; BIC = Schwarz's Bayesian Information Criteria.

Model I is the null model, a baseline model without any determinant variable.

Model II = individual level variables.

Model III = Contextual level variables.

Model IV is the final model adjusted for individual and Contextual level variables.

had no predictors (random intercept). Afterwards, the model II contained only the individual-level variables, model III contained only contextual level variables, while model IV contained both individual level and contextual level variables. For all models, adjusted odds ratios (AOR) and their associated 95% confidence intervals (CIs) were presented. These models were fitted by a STATA command "melogit" for the identification of predictors with the outcome variable. For model comparison, the log-likelihood ratio (LLR) and Akaike information criteria (AIC) test were used. Using the variance inflation factor (VIF), the multicollinearity test showed that there was no evidence of collinearity among the independent variables (Mean VIF = 1.51, Maximum VIF = 2.09 and Minimum VIF = 1.09). Sample weight (v005/1,000,000) was applied in all the analysis to correct for over- and under-sampling while the SVY command was used to account for the complex survey design and generalizability of the findings.

### Ethical approval

Ethical clearance was obtained from the Ethics Committee of ORC Macro Inc. as well as Ethics Boards of partner organisations of the various countries, such as the Ministries of Health. The DHS follows the standards for ensuring the protection of respondents' privacy. Inner City Fund International ensures that the survey complies with the U.S. Department of Health and Human Services regulations for the respect of human subjects. The survey also reports that both verbal and written informed consent were obtained from the respondents. However, this was a secondary analysis of data and, therefore, no further approval was required for this study. Further information about the DHS data usage and ethical standards are available at http://goo.gl/ny8T6X.

## Results

### Prevalence of barriers to healthcare access

Figs 1 and 2 show the prevalence of barriers to healthcare access among women in SSA. From Fig 1, 61.5% of the women had at least one barrier in accessing healthcare. This ranged from 36.3% in South Africa to 84.4% in Chad. The major barrier these women faced was getting money needed for treatment (50.1%) and the least was getting permission to go (15.9%) (see Fig 2).

### Socio-demographic characteristics and barriers to healthcare access among women in SSA

Table 1 shows the socio-demographic characteristics and barriers to healthcare access among women in SSA. About 20% of the respondents were aged 15–19. More than half (54.8%) were married, 32.3% had primary level of education, 32.6% were not working, and 39.1% had 1–3 children. The greater percentage of the women were not covered by health insurance (91.5%). With access to mass media, 38.8%, 77.5%, and 58.5% were not exposed to radio, newspaper, and television respectively. The majority (72%) were in male-headed households, 24.1% were in the richest wealth quintile, and 62.1% were in rural areas. The Chi-square analysis showed that all the independent variables are associated with barriers to healthcare accessibility at p<0.05.

### Factors associated with barriers to healthcare access among women in SSA

Table 3 presents results on the factors associated with barriers in healthcare access among women in SSA. In terms of age, the result showed that women age 35–39 had the lowest odds in barriers to healthcare accessibility (AOR = 0.945, CI: 0.911–0.980), compared to those aged 45–49. In terms of marital status, married women had lower odds of facing barriers in healthcare accessibility (AOR = 0.694, CI: 0.658–0.732), compared to women who were widowed. Compared with women with higher level of education, those with no formal education had highest odds of facing barriers to healthcare accessibility (AOR = 1.803, CI:1.718–1.891). Regarding employment status, compared to those engaged in clerical works, those who are not working (AOR = 1.449, CI: 1.334–1.573), those engaged in agriculture (AOR = 1.909, CI = 1.755–2.075), and manual workers (AOR = 1.551, CI: 1.424–1.689) had higher odds of facing barriers to healthcare. In relation to parity, those with parity 4 or more [AOR = 1.211, CI: 1.169–1.255] had higher odds of facing barriers to healthcare. The result also showed that those who were not covered by health insurance had higher odds (AOR = 1.284, CI: 1.248–1.322) of barriers to healthcare accessibility, compared to those who were covered by health insurance. The result also showed that those who watched television at least once a week (AOR = 0.598, CI:0.571–0.626) and those read newspaper or magazine at least once a week (AOR = 0.893, CI:0.811–0.983) had lower odds of healthcare accessibility barriers, compared

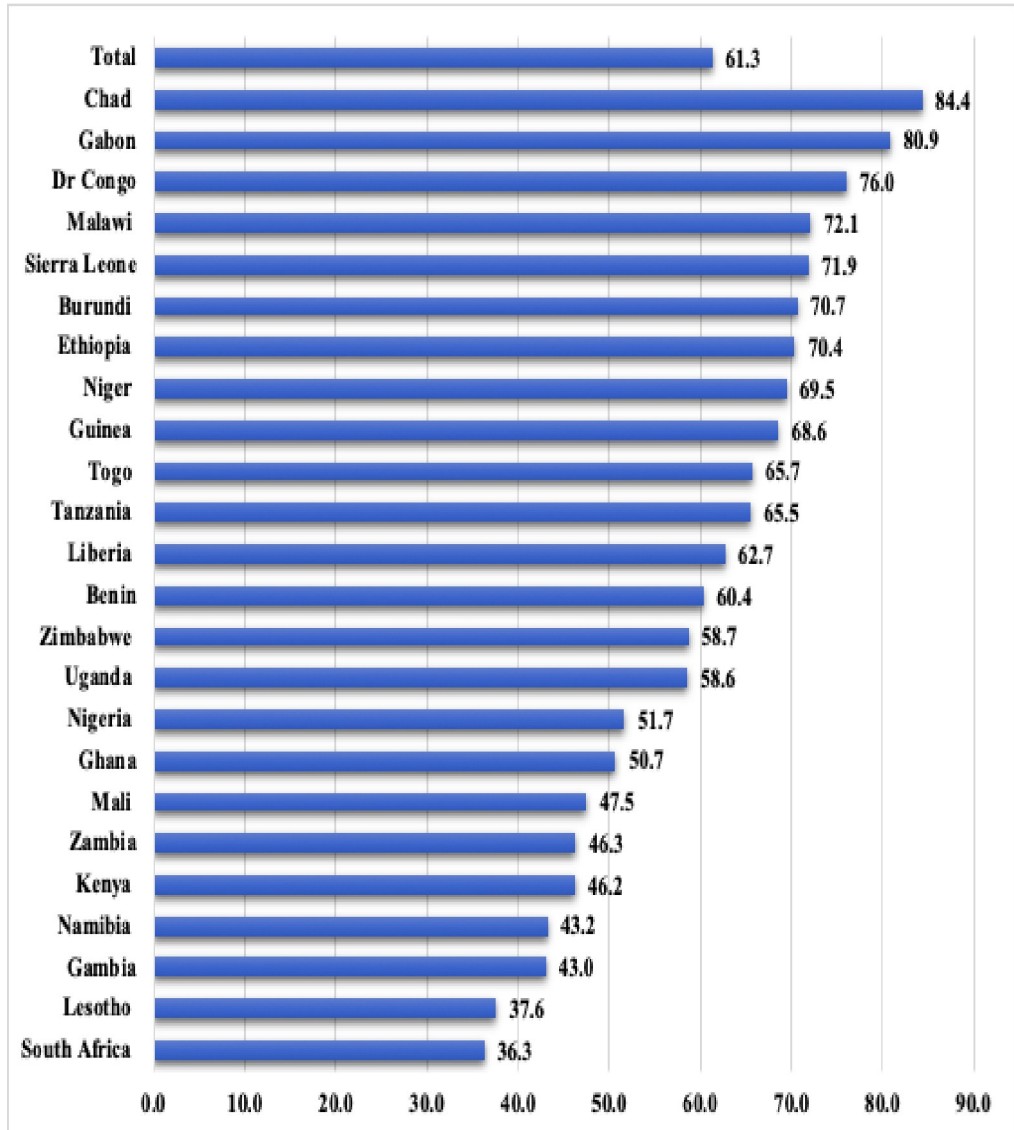

**Fig 1. Prevalence of barriers to healthcare access among women in SSA (%).**

to those who watched television and read newspaper or magazine almost every day. With the contextual factors that were considered in the study, the result also showed that women in the richest wealth quantile had lower odds of facing barriers to healthcare, compared to women in the poorer wealth quantile (AOR = 0.457, CI:0.443–0.472). Those in rural areas (AOR = 1.235, CI:1.209–1.262) had higher odds of facing barriers to healthcare, compared with those in urban areas.

## Discussion

### Summary of main findings

This study sought to assess the individual and contextual factors associated with barriers to healthcare among women in SSA. The results showed that barriers to accessing healthcare is

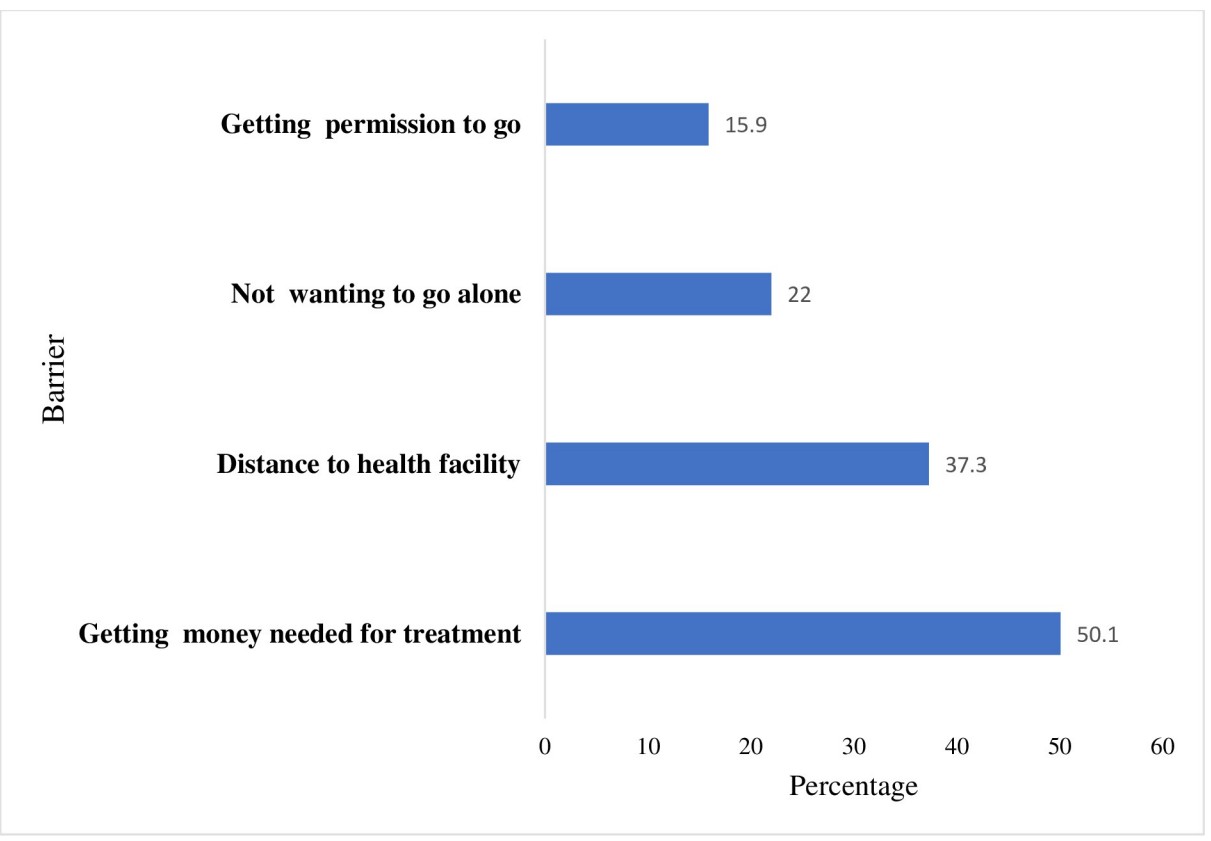

**Fig 2. Types of barriers faced.**

prevalent among women in SSA, with every 6 out of 10 women facing barriers in accessing healthcare. Getting money needed for healthcare and distance to healthcare are the major barriers. The individual factors associated with barriers to accessing healthcare are age, marital status, level of education, employment, parity, health insurance subscription, frequency of listening to radio, frequency of reading newspaper or magazine, and frequency of watching television. The contextual factors associated with barriers to healthcare are wealth status and place of residence.

## Synthesis with previous evidence

The prevalence in this study is similar to prevalence of healthcare accessibility barriers reported in South Africa (65%) [20], Rwanda (64%) [18], Ethiopia (69%) [17], and Tanzania (65%) [16]. The predominant barriers were getting money needed for treatment and distance to health facility. This confirms a previous study in Ethiopia by Tessema and Kebede [17]. The study also showed that women aged 30–34 and 35–39 had lower odds of facing barriers in accessing healthcare, compared to those aged 45 and above. This finding corroborates what has been observed in previous studies in other countries such as Nigeria [21] and Malaysia [22].

The study found that married women and those who had never married married had lower odds of facing a barrier to healthcare access, compared to the widowed. This confirms the findings of several empirical studies in other countries such as Southern Ethiopia [23], Tanzania [16], Afar Region of Ethiopia [24], Montenegro [25], and Malaysia [26].The probable

explanation is that married women may gain economic and psychosocial support from their spouses to access healthcare [27].With the widowed women, in some parts of SSA, certain socio-cultural practices and customs deny them of befitting inheritance, social protection, and access to healthcare. Azah [28], for instance, opined that some widowhood rites in Ghana usually lead to women's inability to inherit their partners' property, which leaves them in abject poverty, marginalized, and unable to afford healthcare. Govender and Penn-Kekana [29] similarly alluded to the fact that unfavourable socio-cultural practices towards widows in many low- and middle-income countries inhibit them from accessing healthcare.

The study also showed that women with no formal education and those with low level of education had higher odds of experiencing barriers to healthcare. Relatedly, the study established that wealth as a contextual factor was also a significant determinant of barrier to healthcare access. Specifically, those in the richest wealth quintile had lowest odds of barrier to healthcare access. In addition, unemployed women had higher odds of facing barriers to healthcare access. Similar findings were reported in previous studies in Ghana [30], Tanzania [16], Uganda [31], Afghanistan [32], Ethiopia [10, 24], and Southern Mozambique [33]. Wealth, education, and employment are proxy measures of socio-economic status, which has been found to be associated with access to healthcare. Specifically, those in high socio-economic status may be in a better position to afford the cost associated with accessing healthcare, which is a common challenge among poorer women [34].The highly educated women are also likely to be in higher paid jobs and, as such, could easily afford healthcare no matter the cost and distance. The highly educated women, all things being equal, are also more informed regarding their fundamental human rights and may have higher health literacy. As a result, they are more likely to overcome any form of barrier to healthcare, compared to their counterparts who are less educated and may have lower health literacy, which has been found to be a key barrier to healthcare utilization [34]. High education and good job may give women the financial power and independence to enable them to afford healthcare, thereby overcoming the barrier of cost, distance, and decision-making [35].

Another key finding in this study is that women who were not covered by health insurance were more likely to face barriers in accessing healthcare. Theoretically, this finding could be argued within the context of the healthcare utilization model by Anderson and Newman [36], which stipulates that health insurance subscription is an enabling factor to healthcare accessibility. The finding also supports findings from previous studies in Ghana [37–39] which showed that health insurance ownership facilitates access to various maternal healthcare services.

It was also found that women who reside in rural areas had higher odds of barriers of healthcare access, compared to urban dwellers. This is in line with other studies in Ghana [30], Tanzania [16], and South Africa [20] which also found a higher likelihood of barriers to healthcare access in rural areas. The basic explanation could be that, in most parts of SSA, rural areas are less privileged manifesting in less health infrastructure, bad road network, and influence of socio-cultural practices that demand women to seek permission from their partners before seeking healthcare [17].

Exposure to mass media also showed decreased odds of healthcare accessibility barriers, which corroborates earlier studies in Ethiopia [40], India [41], Bangladesh [42], and rural Malawi [43]. The reason for this could be that listening to radio, reading newspaper, and watching television increase ones' health literacy, which has been identified as a key enabler to healthcare utilization [44].

## Strengths and limitations of the study

The key strength of this study is the use of nationally representative data to assess individual and contextual factors associated with barriers to accessing healthcare among women in SSA. The findings can, therefore, be generalized to all women in their reproductive age in SSA. The study also employed advanced statistical models, which accounted for the clusters within the sample. Despite these strengths enumerated, the study design was cross-sectional and, therefore, causal interpretation cannot be deduced. Finally, due to the fact that secondary data was used, health-worker related factors could not be accounted for in this study.

## Conclusion

It was found that 61.5% of women face barriers in accessing healthcare in SSA. The major barriers were getting money needed for treatment and distance to health facility. Both individual and contextual factors were associated with barriers to healthcare accessibility. Particularly, age, marital status, employment, parity, health insurance coverage, frequency of listening to radio, frequency of reading newspaper or magazine, frequency of watching television, wealth status, and place of residence were associated with barriers to healthcare accessibility. These factors ought to be considered at the various countries in SSA to strengthen existing strategies and develop new interventions to help mitigate barriers to accessing healthcare among women. Specifically, some of the SSA African countries can adapt successful programs in other SSA countries to suit their context such as the National Health Insurance Scheme (NHIS) and the Community Health-based Planning Services (CHPs) concept in Ghana. There is also the need to empower women economically. These will aid in the achievement of the SDGs, 3.1, 3.7, and 3.8.

## Acknowledgments

I am grateful to MEASURE DHS project for giving me free access to the original data. I also extend my gratitude to Mr. Ebenezer Agbaglo of the Department of English of the University of Cape Coast for copyediting this manuscript for me.

## Author Contributions

**Conceptualization:** Abdul-Aziz Seidu.

**Data curation:** Abdul-Aziz Seidu.

**Formal analysis:** Abdul-Aziz Seidu.

**Funding acquisition:** Abdul-Aziz Seidu.

**Investigation:** Abdul-Aziz Seidu.

**Methodology:** Abdul-Aziz Seidu.

**Project administration:** Abdul-Aziz Seidu.

**Resources:** Abdul-Aziz Seidu.

**Software:** Abdul-Aziz Seidu.

**Supervision:** Abdul-Aziz Seidu.

**Validation:** Abdul-Aziz Seidu.

**Visualization:** Abdul-Aziz Seidu.

**Writing – original draft:** Abdul-Aziz Seidu.

**Writing – review & editing:** Abdul-Aziz Seidu.

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
