## [Decision Letter · Decision Letter 0]

15 Oct 2020

Mixed effects analysis of factors associated with barriers to accessing healthcare among women in sub-Saharan Africa: insights from demographic and health surveys

PONE-D-20-26681

Dear Dr. Abdul-Aziz Seidu,

We’re pleased to inform you that your manuscript has been judged scientifically suitable for publication and will be formally accepted for publication once it meets all outstanding technical requirements.

Kind regards,

Yuka Kotozaki

Academic Editor

PLOS ONE

Journal Requirements:

1. Your ethics statement should only appear in the Methods section of your manuscript. If your ethics statement is written in any section besides the Methods, please delete it from any other section.

Additional Editor Comments (optional):

Reviewers' comments:

Reviewer's Responses to Questions

**Comments to the Author**

1. Is the manuscript technically sound, and do the data support the conclusions?

Reviewer #1: Yes

2. Has the statistical analysis been performed appropriately and rigorously? 

Reviewer #1: Yes

3. Have the authors made all data underlying the findings in their manuscript fully available?

Reviewer #1: No

4. Is the manuscript presented in an intelligible fashion and written in standard English?

Reviewer #1: Yes

5. Review Comments to the Author

Reviewer #1: This is a brief but useful paper that reports high quality data from representative surveys of women in 24 sub-Saharan nations that were conducted between 2010-2018. A useful literature review is presented. All data collection was covered by ethical review. Analyses of the merged, 24-nation data set appear to have been conducted appropriately, with adjustments introduced for data weights and clustering. Findings make good sense and provide valuable important documentation regarding healthcare access on the continent. Limitations are also acknowledged. I find little to criticize or recommend in the way of improvement, although it is unfortunate that the data set is not freely available to other researchers.

6. PLOS authors have the option to publish the peer review history of their article (what does this mean?). If published, this will include your full peer review and any attached files.

Reviewer #1: No

---

## [Editor Report · Acceptance letter]

23 Oct 2020

PONE-D-20-26681 

Mixed effects analysis of factors associated with barriers to accessing healthcare among women in sub-Saharan Africa: insights from demographic and health surveys 

Dear Dr. Seidu:

I'm pleased to inform you that your manuscript has been deemed suitable for publication in PLOS ONE. Congratulations! Your manuscript is now with our production department. 

Kind regards, 

on behalf of

Dr. Yuka Kotozaki 

Academic Editor

PLOS ONE